# Study on the Soil Deterioration Mechanism in the Subsidence Zone of the Wildcat Landslide in the Three Gorges Reservoir Area

**Ruihong Wang [1,2,†], Kaiqiang Zhao [2,†], Can Wei [2,*,†], Xuan Li [2,†], Mingxin Li [2,†] and Jianfeng Zhang [2,†]**

1   Key Laboratory of Geological Hazards on Three Gorges Reservoir Area, Ministry of Education, China Three Gorges University, Yichang 443002, China
2   College of Civil Engineering and Architecture, China Three Gorges University, Yichang 443002, China
*   Correspondence: wc1995@ctgu.edu.cn; Tel.: +86-1816-239-2892
†   These authors contributed equally to this work.

**Abstract:** The stability of soil mass near the dam bank in the Three Gorges Reservoir is closely related to the periodic variation in the reservoir water level. In order to study the influence of water level variation on soil mass, the soil mass in the water level fluctuation zone of the Wildcat landslide was taken as the research object, and the mechanism of soil mass deterioration in this area was revealed by comparing the strength and mineral structure characteristics of soil mass at different elevations by means of macro- and meso-microscopic analysis. The results show the following: (1) With the increase in sampling elevation, the natural water content of the soil decreases, and the dry density of the soil is a minimum when the elevation is 155 m and at a maximum when the elevation is 175 m. (2) The soil mass in the water dissipation zone of the Wildcat landslide can be divided into three areas: When the elevation is 145–155 m, the fractal dimension increases, the soil fragmentation increases, the cohesion decreases, and the soil deterioration increases. When the elevation is 155–175 m, the fractal dimension decreases, the soil fragmentation decreases, the cohesion increases, and the soil deterioration weakens. When the elevation is greater than 175 m, there is no soil deterioration. (3) X-ray diffraction (XRD) and nuclear magnetic resonance(NMR) were used to test the soil's mineral composition and pore size at different elevations. It was found that the main reason for the severe deterioration of macro-strength parameters of the soil at the elevation of 155 m was that the proportion of clay minerals and quartz was at the lowest, and the proportion of medium pores and large pores was at the highest. (4) Through the combination of macro and mesoscopic testing and analysis, it was found that the rise and fall of the reservoir water level will lead to the strong chemical action of the skeleton and cemented mineral dissolution in the soil degradation-enhanced area, as well as the physical action of soil particles, resulting in the formation of more medium pores and large pores in the soil and eventually the formation of seepage channels.

**Keywords:** wildcat face slide; hydro-fluctuation belt; different elevation; soil mass; macro and micro tests





## 1. Introduction

With the construction of the Three Gorges Project, the reservoir began to fill with water in 2003. Since 2008, the water level elevation of the reservoir has varied between 145 and 175 m, forming a 30 m water-leveling zone, which leads to the long-term alternating dry and wet environment on the bank slope of the reservoir and speeds up the deterioration process of rock and soil in the water-leveling zone of the reservoir. After years of development, this can induce many geological disasters, such as landslides and collapses. Therefore, it is of great significance to study the deterioration effect of rock and soil mass in the reservoir bank fluctuation zone for a deeper understanding of the geological hazard occurrence mechanism in the Three Gorges Reservoir area.

In recent years, some scholars have gradually begun to pay attention to the dry–wet cycling of rock and soil mass in the water–level zone along the bank slope of the reservoir, and have carried out a series of targeted laboratory test studies. Hale [1], Apollaro [2], Tallini [3], Alt-Epping [4], and Hurowitz [5] studied the deterioration law of the physical and mechanical properties of different types of rocks under water–rock action, respectively. Qi et al. [6] conducted dry–wet cycle tests on red clay and found that the shear strength parameters of red clay decreased with the increase in the number of dry–wet cycles, and the attenuation was most obvious at the first cycle. Deng et al. [7] conducted dry–wet cycle tests on soil in the moisture-dampening zone and found that the degradation range of soil shear strength parameters caused by the first four dry–wet cycles accounted for about 75% of the total range pairs. Zhang et al. [8] conducted dry–wet cycling–cooling and dry-heating experiments on sandstone in the three gorges reservoir area and found that two physical processes (the dry–wet cycle and cold–heat cycle) would degrade sandstone samples, and the coupling effect of the two processes would accelerate the deterioration rate of sandstone samples. Bai et al. [9] conducted dry–wet cycle tests on loess and found that, with the increase in dry–wet cycle times and cycle amplitude, the cohesion and internal friction angle of loess decreased, and the shear strength of loess deteriorated significantly. Wang et al. [10] conducted a periodic penetration test on the gravel soil in the fluctuation zone to simulate the periodic fluctuation of the water level and found that the gravel soil slope was more prone to particle loss under infiltration due to its physical and mechanical properties, thus leading to strength deterioration. Cheng et al. [11] studied the infiltration of fractured soil by conducting dry–wet cycle tests on soil with different amounts of water and found that its permeability increased with the decrease in the initial water content and the increase in dry–wet cycle times. Sun et al. [12] used indoor model tests to simulate the deformation characteristics of the loess slope under the action of alternating wetting and drying, and found that alternating wetting and drying mainly occurred at the shoulder of the slope, accompanied by a large number of small cracks. The soil at the shoulder and foot of the slope was strained and deformed towards the free surface, while the soil in the middle was compressed.

Different mineral compositions and arrangements of soil mass result in the different microscopic pore structures of rock and soil mass, which is a key factor affecting its macroscopic mechanical characteristics [13]. In recent years, X-ray diffraction (XRD) tests, nuclear magnetic resonance (NMR) tests, scanning electron microscopy (SEM) tests, and image processing techniques (DIC) have been mainly applied to test the microstructural characteristics of soil [14–16]. Tiwari et al. [17] drew a triangular correlation graph between the mineral composition and the compression index and expansion index to estimate the compression index and the expansion index of soil. Elhassa et al. [18] selected six different soils for XRD mineral composition analysis when studying road foundation materials and found that kaolinite was significantly more effective than montmorillonite in reducing the plasticity of clay. Zhu et al. [19] used a mercury injection test to study the development mechanism of cracks and micro-pores in expanded soil after coal gangue improvement under the action of wetting and drying cycles, and they found that coal gangue powder could significantly reduce the proportion of micro-pores, cumulative pore volume, and cumulative pore density in expanded soil, thus reducing macro cracks. Dong et al. [20] used nuclear magnetic resonance (NMR) to predict the strength of bentonite after wetting and drying cycles by comparing the pore variation characteristics of bentonite with different times of wetting and drying cycles. Li et al. [21] used nuclear magnetic resonance (NMR) and scanning electron microscopy (SEM) to observe the pore structure of loess after a freeze–thaw cycle and found that the change in pore structure could reflect the change in loess strength after the freeze–thaw cycle. Xue et al. [22] observed slip-zone soil after brine immersion with SEM images, and the SEM test showed that an increase in salt concentration (0–8%) promoted the formation of small aggregates in loess, and further promoted the improvement in shear strength. Ye et al. [23] used scanning electron microscopy (SEM) to test the micromorphological characteristics of loess under alternating wetting and drying

conditions, and found that the size of clay minerals between soil particles decreased, the contact mode between soil particles developed from stable to unstable, and the particles gradually became round. Xie et al. [24] extracted crack parameters from images of red clay after cyclic drying and wetting through image processing technology (DIC) to analyze the crack development law.

At present, most scholars study the influence of reservoir water on soil by changing the test conditions of laboratory tests, but they ignore the difference in soil properties at different elevations caused by the change in water level in the fluctuation zone. Therefore, this paper focuses on an experimental test of soil mass in the water level zone of the Wildcat landslide and reveals the soil mass's deterioration mechanism by comparing its characteristics at different elevations and adopting the analysis method combining macro and meso.

## 2. Materials and Methods

The Wildcat landslide is the large landslide body closest to the dam in the Three Gorges Reservoir area of the Yangtze River. It is located on the north bank of the Yangtze River in Zigui County, 12 km away from Zigui Old Town (Guizhou Town) on the upper side and 17 km away from the dam site of the three gorges project on the lower side, as shown in Figure 1A,B. The Wildcat Landslide is the nearest super-large landslide to the Three Gorges Dam. Figure 1C shows the full picture of Wildcat landslide. After the operation of the Three Gorges Reservoir water level, the elevation of the subsidence zone is 145–175 m, and some signs of deformation adjustment appear in the local landslide. Therefore, the object of this study was the soil of different elevations in the area of the subsidence zone of the Wildcat landslide. The sampling time was in early July. The lowest water level of the Three Gorges Reservoir was 145 m. Field sampling is shown in figure Figure 1D.

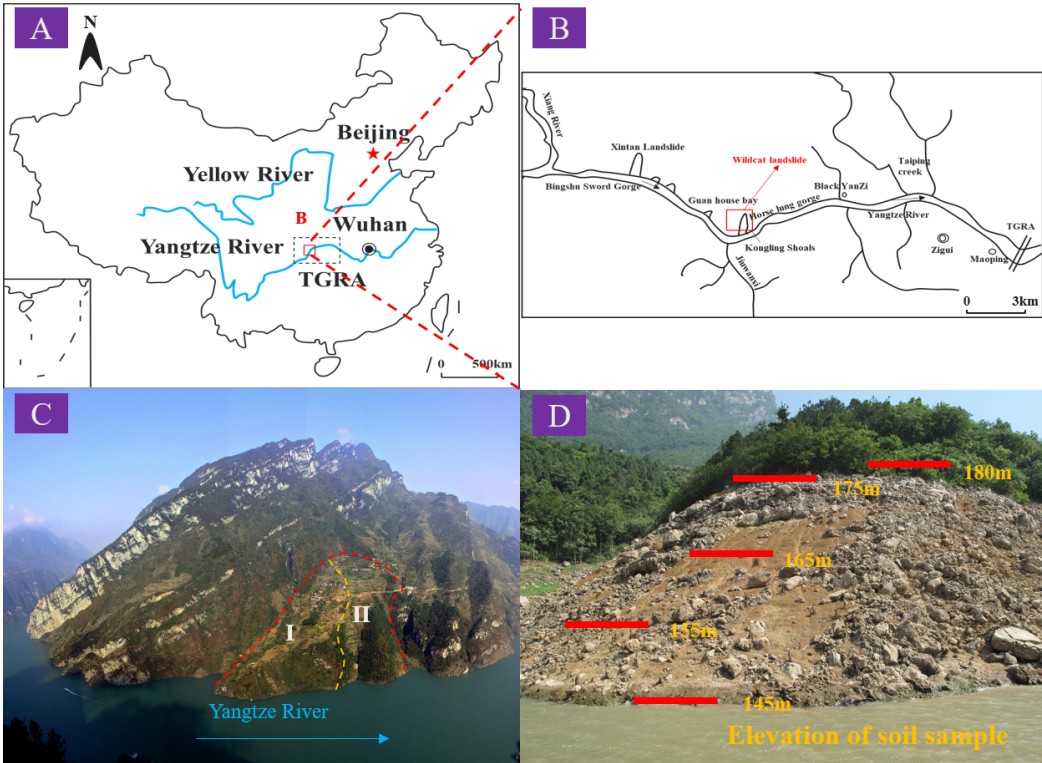

**Figure 1.** Sampling spot. (**A**,**B**) is the position of theWildcat landslid; (**C**) shows the full picture of Wildcat land; field sampling is shown in (**D**).

The test process of this study is shown in Figure 2. (1) When the lowest water level of the Three Gorges Reservoir area was 145 m, soil samples were selected from the wildcat

landslide site at different elevations. The elevations were 145 m, 155 m, 165 m, 175 m and 180 m, and 50 kg soil samples were selected for each elevation. (2) After the soil sample was retrieved, the wax sealing method was used to test the dry density of the soil in strict accordance with the specification GB/T50123-1999 Geotechnical Test Method, as shown in Figure 2A. At the same time, the remaining samples were naturally dried. After drying, the dry soil sample was hammered out with a rubber hammer. (3) Screening tests were conducted on soil at different elevations, as shown in Figure 2B. After screening, the constant head method was used to test the soil permeability coefficient, as shown in Figure 2C. Direct shear tests were conducted on soil samples to test the shear strength of soil, as shown in Figure 2D. (4) X-ray diffraction (XRD) was used to test the mineral composition of soil at different elevations and observe its mineral composition and content, as shown in Figure 2a. (5) Nuclear magnetic resonance (NMR) was used to test the pore characteristics of soil at different elevations, as shown in Figure 2b. Because the periodic change in water level leads to the different characteristics of soil at different elevations, the mechanism of soil deterioration in the subsidence zone can be revealed by using the analysis method combining macro and meso.

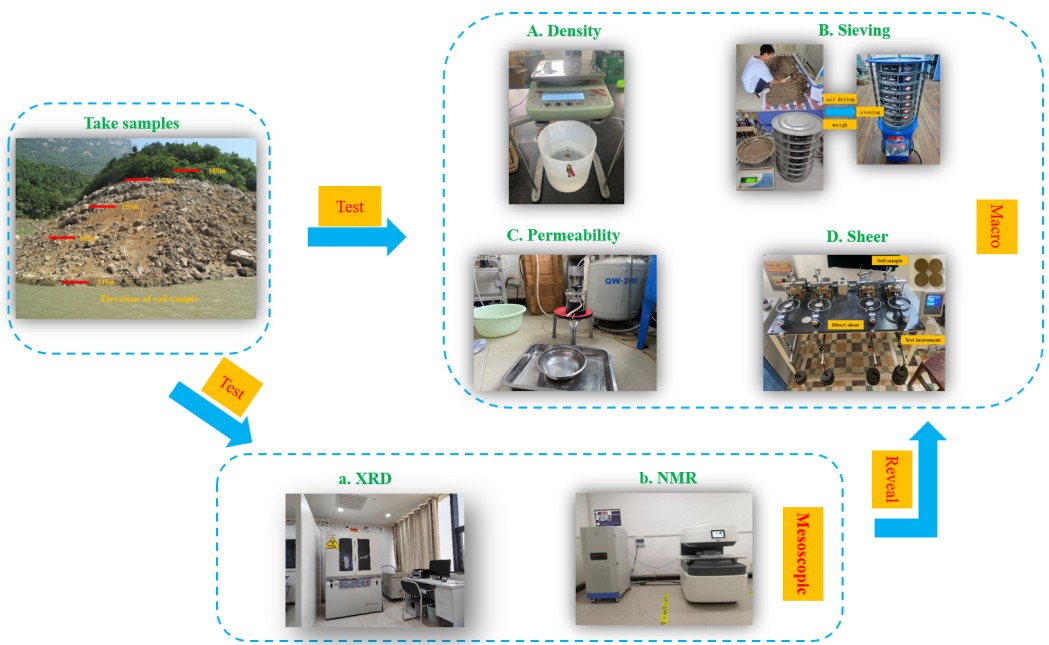

**Figure 2.** Test process diagram.

### 3. Results

*3.1. Physical Index*

Soil samples were selected from five different elevations in the water-leveling zone of the Wildcat landslide, and the natural moisture content of the soil was tested by the drying method. Figure 3 shows the various rules of the dry density and natural water content of the samples of the soil at different elevations. It can be seen in the figure that the natural water content of soil samples is negatively correlated with the sampling elevation and decreases with the elevation increase. After the soil mass was dried for 24 h in the oven, the wax sealing method was used to test the dry density of the soil sample. As shown in Figure 3, the dry density of the soil sample was at a minimum when the sampling elevation was 155 m, and at a maximum when the sampling elevation was 175 m. When the sampling elevation was between 155 m and 175 m, the dry density of the soil sample increased with the increase in the sampling elevation. When the sampling elevation was 180 m, the dry density of the soil sample began to decrease, and when the sampling elevation was 145 m, the dry density of the soil sample was greater than that of the soil sample at the elevation of 155 m.

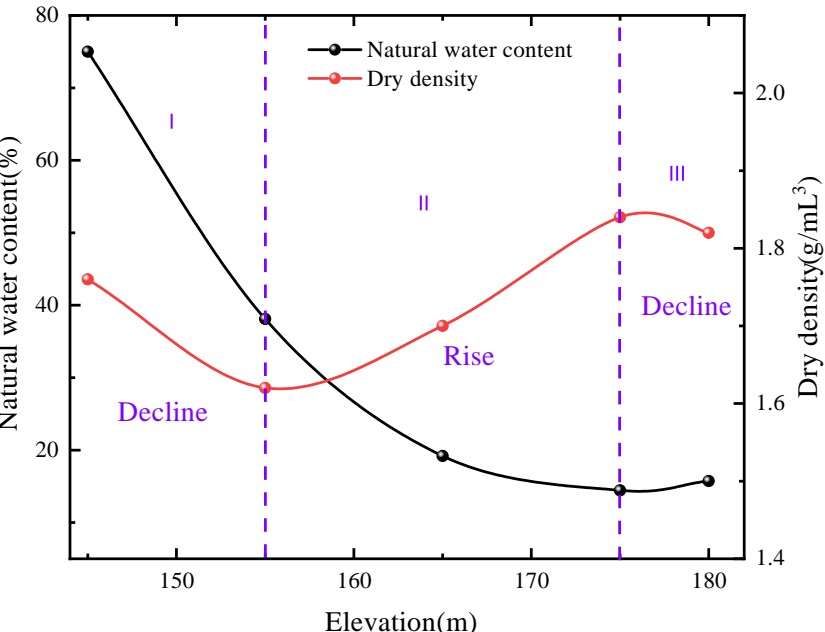

**Figure 3.** Density and natural moisture content.

The above rules show that the natural water content and dry density of the soil at different elevations are closely related to the water level elevation. The closer the soil sample is to the water level elevation of 145 m, the higher the natural water content is and the lower the dry density is. The farther away the soil sample is from that water level elevation, the lower the natural water content is and the higher the dry density is. Therefore, the fluctuation of the water level in the water-leveling zone will lead to the long-term unsaturated state of the soil mass in the region, and the moisture content of the soil mass at different elevations will be different.

The landfall zone of the Wildcat landslide is a soil–rock accumulation landslide, and the soil of different elevations belongs to the same area. The distribution and structure of soil particles in macro and meso are self-similar. Therefore, according to [25], the fractal theory can be used to quantitatively fractal the self-similarity of this kind of soil–rock mixture. In fractal theory, the number $N(R)$ of particles with particle size less than or equal to $R$ has a positive correlation with particle size (a simplified expression of the constant term), namely

$$N(R) = R - D \tag{1}$$

In Equation (1), D is the fractal dimension.

In practice, it is found that when the particle size is large, it can be read; when the particle size is small, the size of the particle group is too large to be obtained directly. Equation (1) can be derived and transformed into a mass distribution model, and the derivation process is as follows:

$$dN(R) = R^{-D-1}dR \tag{2}$$

If the particles are assumed to be spherical, then the sum of the particle mass of a certain particle group $dm(R)$ is proportional to the product of the particle size $R^3$ and the number of particles $dN(R)$:

$$dm(R) = R^3 dN(R) \tag{3}$$

Combined with Equations (2) and (3), we obtain the following:

$$M(R) = R^{3-D} \tag{4}$$

The cumulative mass $M(R)$ of particles smaller than a certain particle size can be obtained by integrating both ends within the particle size range in Equation (4).

$$dm(R) = R^{2-D}dR \tag{5}$$

According to the characteristics of the gradation curve, the cumulative mass percentage $P(R)$ is positively correlated with $M(R)$, so Equation (5) is equivalent to $P(R) = R^{3-D}$, and logarithms are taken at both ends:

$$lg[P(R)] = (3-D)lgR \tag{6}$$

According to Equation (6), $lg[P(R)]$ is taken as the vertical coordinate and $lgR$ as the horizontal coordinate, the skew rate $n$ is calculated by linear fitting, and the fractal dimension of graininess is $D = 3 - n$.

In the indoor laboratory, to retrieve the site soil sample screening test, the mass percentage distribution of soil particles at different elevations in the water dissipation zone of the Wildcat landslide was obtained, and the fractal theory was used to deduce it. Figure 4 shows the fractal curves of the soil at different elevations. The linear fitting curve is shown in Equation (7):

$$y = nx + a \tag{7}$$

According to the linear fitting curve, the fractal dimension of soil mass is calculated, as shown in Table 1. The fitting coefficient is between 0.95 and 0.99, and the fitting effect is ideal. The fractal dimension D of soil is between 2.15 and 2.28.

**Table 1.** Fractal dimension of soil with different elevations.

| Soil Sample Elevation | $n$ | $a$ | $R^2$ | $D = 3 - n$ |
|---|---|---|---|---|
| 145m | 0.80 | 1.24 | 0.99 | 2.20 |
| 155m | 0.72 | 1.25 | 0.98 | 2.28 |
| 165m | 0.74 | 1.46 | 0.95 | 2.26 |
| 175m | 0.85 | 1.46 | 0.95 | 2.15 |
| 180m | 0.76 | 1.35 | 0.98 | 2.24 |

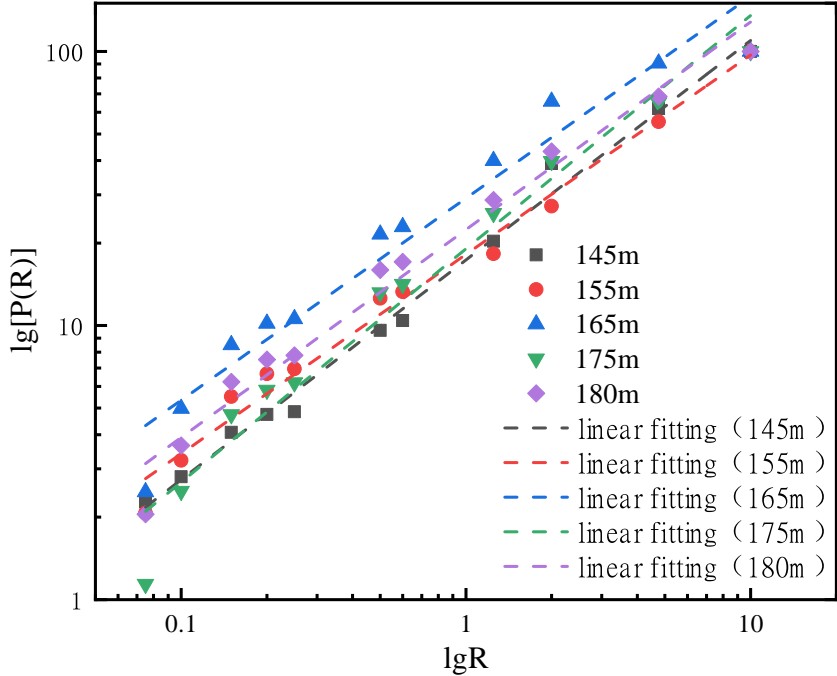

**Figure 4.** Fractal curves of soil with different elevations.

The fractal theory is a comprehensive fractal of all soil particles, which can be regarded as a general understanding of soil particles from local to the whole. Therefore, the fractal dimension can be used as a quantitative characteristic index of soil gradation characteristics. The larger the fractal dimension, the higher the degree of soil particles' fragmentation. When the soil elevation is 155 m, the maximum fractal dimension is 2.28, indicating that the soil fracture degree is the highest. When the soil elevation is 175 m, the fractal dimension is 2.15, indicating that the soil fracture degree is the lowest. Furthermore, although the soil is selected as the same area, the fractal dimension of the soil at different elevations is different, so the deterioration degree of the soil at different elevations is also different.

The permeability coefficient comprehensively reflects the difficulty of water flowing in the pores of the soil, which is mainly affected by the particle size (mineral composition, particle size, and grading), density (expressed by available pore ratio e), humidity (expressed by the available water content/saturation), structure (soil structure) and hydraulic path. In accordance with "GB/T50123-1999 Geotechnical test Method", the saturated permeability coefficient $k$ of the soil at different elevations was tested using the constant head test. The calculation formula of the pore ratio is shown in Equation (8):

$$e = \frac{m_1 - m_2}{m_2} \times 100\% \tag{8}$$

where $e$ is the pore ratio, $m_1$ is the saturated mass of soil, and $m_2$ is the dry mass of soil.

Figure 5 shows the variation law of the soil permeability coefficient and the pore ratio with elevation. It can be seen in the figure that the soil permeability coefficient and porosity ratio change in the same law. When the porosity ratio of soil increases, the permeability coefficient also increases, and when the porosity ratio decreases, the permeability coefficient also decreases. When the sampling elevation is 155 m, the soil pore ratio is the largest, and the permeability coefficient is also the largest. When the sampling elevation is 175 m, the soil porosity ratio is the smallest, and the soil permeability coefficient is the smallest. Therefore, according to the law of water seepage in the soil of different elevations, the soil seepage of different elevations in the fluctuation zone is defined by three regions: Region I, with an elevation of 145–155 m, is the penetration acceleration zone. Region II has an elevation of 155–175 m and is an osmotic deceleration zone. The elevation of Region III is more than 175 m and permeability is lowest at 175 m, but gradually increases with elevation. According to the above rules, due to the periodic changes in reservoir water, the deterioration of the soil mass of 155 m is the most severe. The internal pores of the soil mass are large, and the water flow rate in the soil mass is fast. Therefore, the pore ratio and permeability coefficient of the soil mass at the height of 155 m are the largest.

### 3.2. Strength Characteristics

As shown in Figure 6, the change rules for the soil viscosity and friction angle are different at different sampling elevations. Comparing the sampling elevations of 145 m and 155 m, the soil cohesive force is larger, and the friction angle is smaller at 145 m. When the elevation is 155 m, the soil cohesive force is at a minimum, and the friction angle is at a maximum. When the sampling elevation is greater than 155 m and less than 175 m, the cohesion of the soil increases with the increase in elevation, and the friction angle decreases with the increase in elevation. When the sampling elevation is greater than 175 m, the cohesion of the soil decreases, and the friction angle increases. Therefore, different elevations of the fluctuation zone are divided into three areas: The first region has an elevation of 145–155 m, and the soil strength decreases; the second region has an elevation of 155–175 m, and the soil strength increases; the third region has an elevation of more than 175 m, and the soil strength decreases, but the amplitude is not large. There is no deterioration zone above the fall zone.

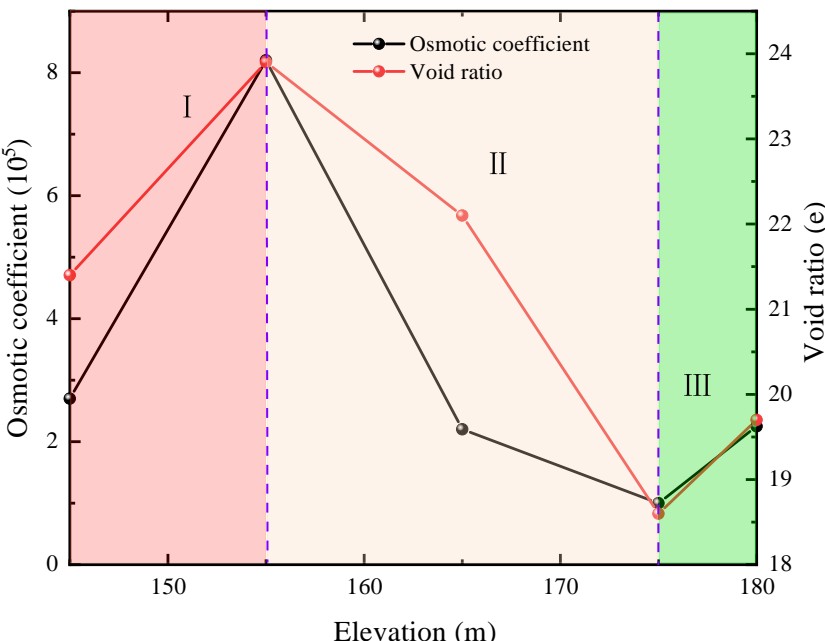

**Figure 5.** Permeability coefficient and pore ratio of soil with different elevations.

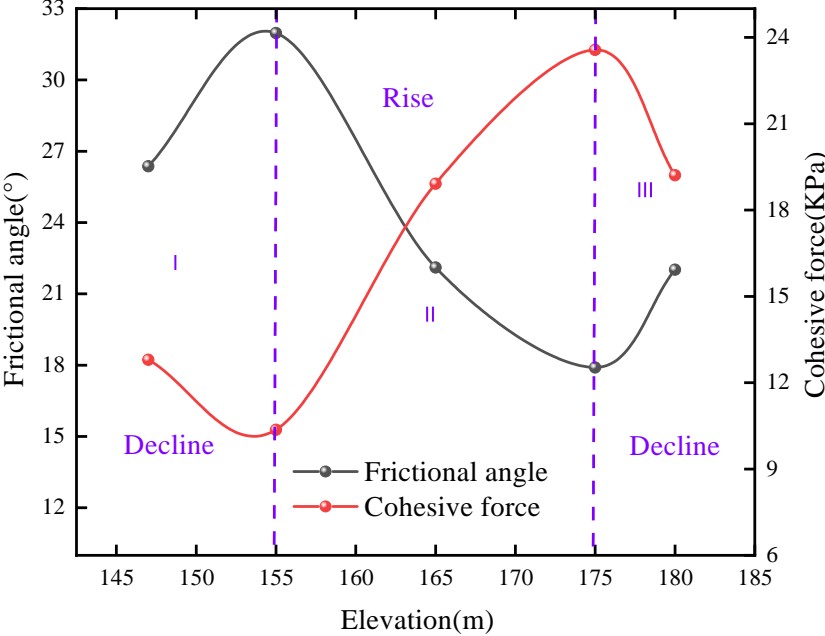

**Figure 6.** Cohesion and friction angle.

Thus, the macroscopic parameters of soil in different elevations at the same sampling location are significantly different. The main reason is that the elevation of the reservoir water level in the water-leveling zone of the Wildcat landslide varies from 145 to 175 m throughout the year. When the elevation is 145 m, it is the same as the lowest water level of the reservoir water, and the natural water content of the soil is the highest. However, because the soil of the Wildcat landslide is mainly gravel soil, under the condition of a water level change, more small gravel is deposited at the lower water level. Compared with the elevation of 155 m, the water level of 145 m contains more gravel. Therefore, when the elevation is 145 m, the soil density is higher, the permeability is smaller, and the soil strength is larger. When the sampling elevation is greater than or equal to 155 m and less than 175 m, the soil moisture content gradually decreases, indicating that the degree of soil water loss increases due to the decrease in water level, the soil density gradually increases,

the permeability gradually decreases, and the soil strength gradually increases. When the sampling elevation is greater than 175 m, the soil is in the non-subsidence zone, there is obvious vegetation growth, and the vegetation roots have the function of water fixation. The soil contains vegetation roots during sampling, but the soil is loose after the removal of the roots in the laboratory test. Therefore, when the sampling elevation is greater than 175 m, the moisture content of the soil increases, the density decreases, the permeability increases, and the soil strength decreases. To sum up, periodic water level variation in the water-leveling zone will lead to significant differences in the physical properties of soil mass at different elevations, leading to different degrees of deterioration of the soil mass.

### 3.3. Microscopic Observations

XRD tests were conducted on the samples of the soil at different elevations, and the test analysis employed samples of soil at an elevation of 145 m as an example, as shown in Figure 7. It can be seen in the figure that the soil mass of the Wildcat landslide is mainly composed of quartz, clay minerals, potassium feldspar, alacrite, calcite, dolomite, tremolite, and other minerals, among which quartz and clay minerals are the main components, accounting for the largest proportion. Figure 8 shows the mineral composition of the soil at different elevations. A comparison shows that the mineral composition of the soil at different elevations is different. When the elevation is 175 m, the total proportion of quartz and clay minerals in the soil reaches 89.67%. When the elevation is 155 m, this proportion is the lowest, only 62.53%. When the elevation is 145 m, this proportion is 81.84%. When the elevation is 165 m, the proportion is 75.76%. When the elevation is 180 m, the proportion is 88.87%.

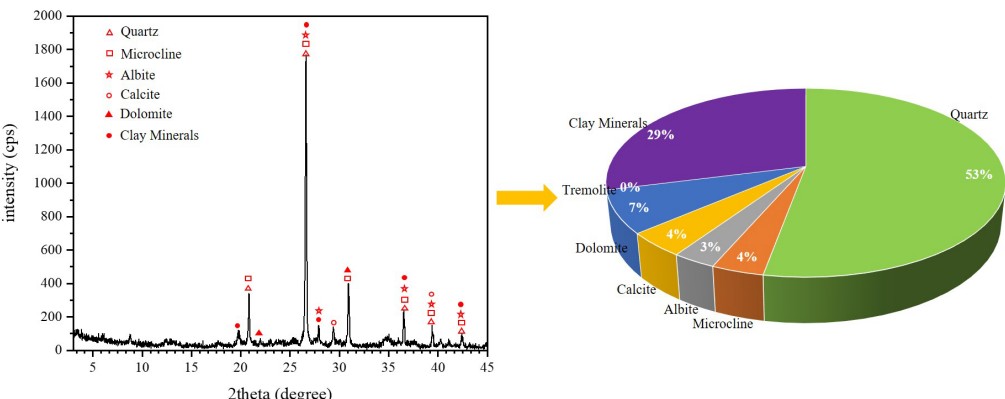

**Figure 7.** XRD test analysis of 145m elevation soil.

The above analysis shows that the largest difference in the mineral composition of the soil at different elevations in the subsidence zone is the difference in the content of clay minerals and quartz. The clay mineral is the main component of the soil cement, and the quartz is the main component of the soil skeleton mineral. Therefore, the different amounts of quartz and clay minerals greatly affect the macroscopic strength properties of the soil. When the soil elevation is 175 m, the content of clay minerals and quartz is the highest, the soil is the most stable, and the cohesion is the highest. When the soil elevation is 155 m, the content of clay minerals and quartz is the lowest, the soil is the most unstable, and the cohesion is the lowest. According to the mineral composition of the soil at different elevations, the soil at different elevations in the subsidence zone can also be divided into three regions: Region I has an elevation of 145–155 m, and the proportion of clay minerals and quartz decreases. The elevation of Region II is 155–175 m, and the proportion of clay minerals and quartz increases. The elevation of Region III is more than 175 m, and the proportion of clay minerals and quartz decreases, but the decreasing range is relatively low.

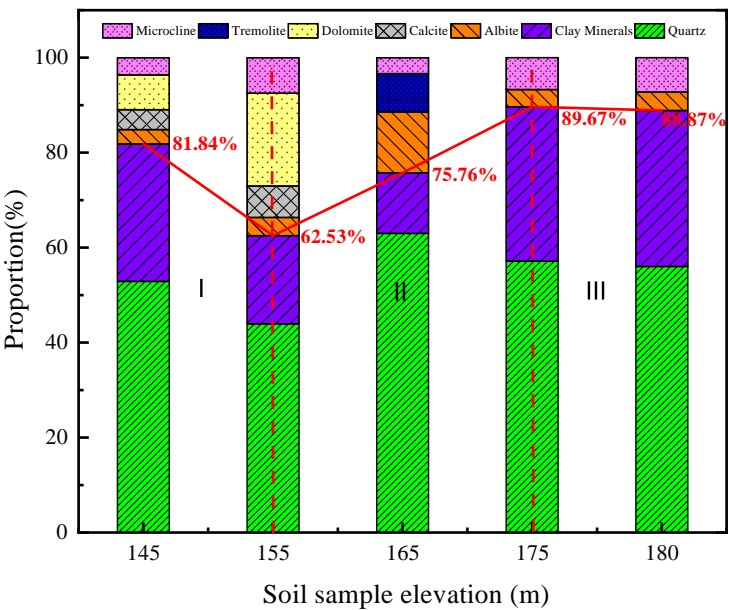

**Figure 8.** Mineral composition of soil at different elevations.

A MacroMR12-150H-1 NMR analyzer was used to test the samples after cyclic loading and unloading, and the pore size characteristics of the soil at different elevations were obtained. According to the dry density of the soil mass, the corresponding volume mass was weighed, and the soil mass was pressed into a PVC pipe with a diameter of 50 mm and a height of 20 mm. The PVC pipe was used to prevent the interference of nuclear magnetic signals, and it was used in conjunction with the nuclear magnetic resonance testing equipment. The pore size distribution was tested using core analysis software, and the pore sizes were defined as different pore types, i.e., small pores (0–1 µm), medium pores (1–10 µm), and large pores (10–25 µm). Through calculation, the proportion of different types of pores is obtained, and the test and calculation results are shown in Figure 9.

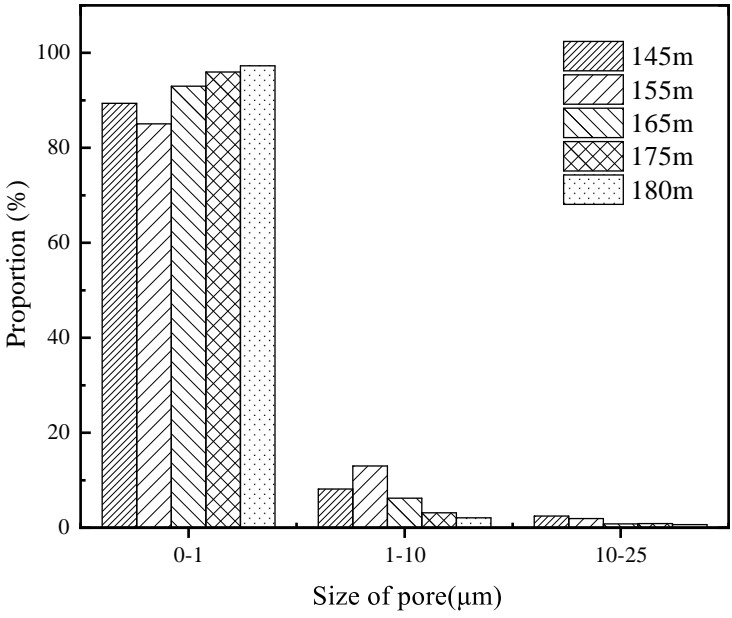

**Figure 9.** Pore size ratio of soil at different elevations.

Figure 9 shows that the soil mass at different elevations contains small pores, medium pores, and large pores. The small pores account for the highest proportion, which is above 85%, and the large pores account for the lowest proportion, which is all less than 10%.

However, comparing the proportion of small pores in the soil at different elevations shows that the proportion of small pores in the soil at an elevation of 155 m is the lowest compared with the soil at other elevations. The proportion of small pores in the soil at an elevation of 180 m is the highest, and the proportion of small pores increases with the increase in sampling elevation between 155 and 180 m. When the soil elevation is 155 m, the proportion of pores is the highest; when the sampling elevation is 180 m, this proportion is the lowest. When the soil elevation is 155–180 m, it decreases with the increase in sampling elevation. The proportion of large pores in the soil at different elevations decreases with the increase in elevation. When the soil elevation is 145 m, this proportion is the highest, and when the soil elevation is 180 m, it is the lowest.

In summary, when the soil elevation is 155 m, the proportion of medium pores and large pores in the soil is much higher than that at other soil elevations, and the proportion of small pores is smaller than that at other soil elevations. The reservoir water permeates into the soil at an elevation of 155 m, forming a large seepage channel, with a large pore size and obvious damage to the soil. Therefore, the permeability coefficient of the high-rise soil is high, and the strength of the soil is low. Regarding soil at a higher elevation, its internal small pores account for a higher proportion, the seepage channel is smaller, the seepage velocity is smaller, and the permeability coefficient is relatively lower. Therefore, its macro strength is higher than that of soil at an elevation of 155 m.

## 4. Discussion

Through the above test analysis of soil characteristics in different regions of the water–leveling zone, it was found that since the Three Gorges Reservoir area was filled with water, the water level has maintained a periodic operation of 145–175 m over the entire year, and the deterioration degree of the soil is different at different elevations. The soil in Region I, with an elevation of 145–155 m, has the lowest dry density, the highest permeability coefficient, the lowest cohesion, and the highest deterioration of soil. The farther away the other areas are from the reservoir water level, the smaller the soil deterioration. Therefore, the soil experienced periodic changes in the reservoir water level, and the soil showed obvious deterioration. The deterioration process is summarized in Figure 10.

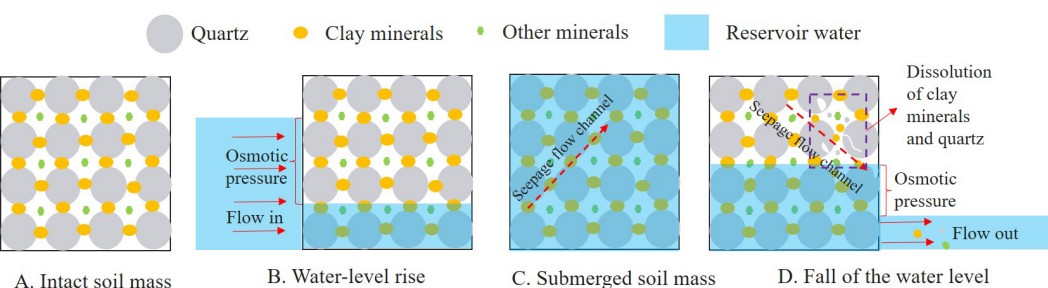

**Figure 10.** Generalized diagram of soil degradation of water level lifting.

As shown in Figure 10A, the intact soil is mainly composed of quartz and clay minerals, and contains a small amount of other minerals. There are micropores among the minerals. When the water level rises, as shown in Figure 10B, the dry soil and the external moist environment cause seepage pressure into the soil, and the water enters the soil, causing the soil particles to become distributed along the water flow direction until all pores between the mineral particles are saturated, as shown in Figure 10C. When the water level drops, as shown in Figure 10D, the external environment of the soil is dry, and the interior of the soil is wet. Seepage pressure will be generated on the outside of the soil, and water will be discharged on the outside of soil particles, resulting in the movement of the soil particles along the water flow direction. Under the condition of a long-term periodic water level change, the soil particles on the bank slope of the reservoir will form seepage channels upon the infiltration of water, resulting in the enhanced permeability of the soil. The number of

pores in the soil particles increases, and the strength of the soil decreases, which eventually leads to the deterioration of the soil.

At the same time, when the soil is in contact with the reservoir water, a water chemical reaction will occur, as shown in Figure 10D. Liquid water is a weak electrolyte and will normally electrolyze small amounts of hydrogen and hydroxide ions, as shown in Equation (9). This reaction equation is a reversible reaction. When the water level drops, the temperature of the soil increases due to solar illumination, and the electrolysis of the water in the soil is accelerated. The content of free water hydrogen ions in the soil increases, the quartz reacts with the cement in the soil, and the quartz dissolves, as shown in Equation (10). At the same time, hydrogen ions react with potassium feldspar and alacrite in the soil to form kaolinite, a clay mineral. The reaction equation is shown in the Equations (11) and (12). However, when feldspar reacts with hydrogen ions, more free potassium ions are generated in the water. The content of the skeleton mineral quartz in the soil is higher, the reverse reaction is enhanced, and more feldspar is generated by the dissolution of clay minerals. Therefore, the long-term rise and fall of the water level will lead to the enhancement of the water chemical reaction of the soil in Region I, as well as the maximum dissolution of quartz and clay minerals in the soil, so the content of the soil is the smallest, the soil particles are broken most severely, and the strength is the lowest. In Region II, the soil elevation is 155–175 m. During the process of water level decline, the soil is in a long-term dry environment; the natural water content of the soil is low, the water chemical reaction is weaker, the amount of clay minerals and quartz in the native soil is higher, the soil particles are more complete, and the strength is higher. In Region III, with an elevation of more than 175 m, the soil chemical reaction is weaker, the soil deterioration is lower, and the strength is higher.

Through the combination of macro and meso test analysis, it is found that the main reason for the different deterioration degrees of soil at different elevations is that soil at different elevations experiences different degrees of water–soil physical and chemical effects. The closer the elevation is to the reservoir water level, the stronger the physical and chemical effects are experienced. Therefore, it is of great significance for landslide prevention and prediction to fully understand the macroscopic mechanical characteristics and meso-structural characteristics of soil mass in the subsidence zone. This study focuses on soil mass of different elevations in the same region. In the future, the research group will compare soil mass of the same elevation in different regions to study the deformation and failure mechanism of soil mass, and to reveal the relationship between soil mass failure in the subsidence zone and the deformation and instability of the landslide body.

$$H_2O \Leftrightarrow H^+ + OH^- \tag{9}$$

$$SiO_2 + 2H_2O \Leftrightarrow H_4SiO_4 \tag{10}$$

$$2K[AlSi_3O_8] + 2H^+ + H_2O \Leftrightarrow 2K^+ + 4SiO_2 + Al2[Si_2O_5][OH]_4 \downarrow \tag{11}$$

$$2Na[AlSi_3O_8] + 2H^+ + H_2O \Leftrightarrow 2K^+ + 4SiO_2 + Al2[Si_2O_5][OH]_4 \downarrow \tag{12}$$

## 5. Conclusions

The purpose of this study was to study the effect of periodic variations in water level on landslide soil. Taking the soil mass in the water level zone of the Wildcat landslide as the research object, the deterioration mechanism of the soil mass in the water level zone of the Wildcat landslide was revealed through analysis of the macroscopic parameter variation rule and the meso-structural characteristics of the soil mass at different elevations. The main conclusions are as follows:

(1) The testing and analysis of the natural water content and dry density of soil at different elevations showed that the natural water content is negatively correlated with

the sampling elevation. The dry density of the soil is at a minimum when the elevation is 155 m and at a maximum when the elevation is 175 m.

(2) Fractal theory was used to calculate the fractal dimension of the soil at different elevations. When the elevation was 155 m, the fractal dimension of the soil was the largest, and the soil particles were broken most severely. Combined with the strength characteristics of the soil mass, the soil mass can be divided by the deterioration degree at different elevations: deterioration enhancement, deterioration weakening, and no deterioration.

(3) A mineral test and nuclear magnetic resonance test conducted on the soil in different areas showed that, when the soil elevation was between 145 and 155 m, the amount of clay minerals and quartz in the soil was the lowest, and there were more internal pores in the soil.

(4) The periodic circulation of the water level will lead to the dissolution of clay minerals and quartz in the soil, and the number of internal pores in the soil will increase. Moreover, the periodic fluctuation of the water level will lead to the formation of seepage channels in the soil, and the macro strength of the soil will decrease.

**Author Contributions:** Data curation, R.W. and C.W.; investigation, K.Z., X.L., M.L. and J.Z.; methodology, R.W. and C.W.; writing—original draft, R.W., K.Z. and C.W.; writing—review and editing, R.W., and C.W. All authors have read and agreed to the published version of the manuscript.

**Funding:** This research was funded by the National Natural Science Foundation of China (grant number 51979151); the Natural Science Foundation of Hubei Province Outstanding Youth Project (Grant number 2021CFA090); and the Three Gorges Key Laboratory of Geological Hazards of the Ministry of Education (China Three Gorges University) (Grant number 2020KDZ07).

**Institutional Review Board Statement:** Not applicable.

**Informed Consent Statement:** Not applicable.

**Data Availability Statement:** Not applicable.

**Acknowledgments:** The authors gratefully acknowledge the Yangtze River Scientific Research Institute for its help in the field sampling process.

**Conflicts of Interest:** The authors declare no conflict of interest.

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
