# Peer review of "Study on the Soil Deterioration Mechanism in the Subsidence Zone of the Wildcat Landslide in the Three Gorges Reservoir Area"

_water, doi:10.3390/w15050886_

Round 1

Reviewer 1 Report

In this paper, the degradation mechanism of soil mass in the landslide subsidence zone is revealed by field sampling, laboratory testing and comparing the macro and micro physical and chemical characteristics of different high-rise soils. It has novel conception, clear thinking and practical engineering significance. In general, the paper is well organized and easy to follow. The manuscript should be accepted for publication on the condition that all the following issues should be addressed.

1. The author should explain the purpose of this study more clearly in the abstract.

2. The author should introduce the innovation of his research content in more detail in the introduction.

3. The calculation result in the table of 175m high level fractal dimension in paper 3.1 is inconsistent with that in the paper. Please check it carefully.

4. The author should explain the variation law of permeability coefficient and pore ratio in paper 3.1.

5. The authors should carefully check whether the pore size unit "μm" in paper 3.3 is misexpressed.

6. In paper 3.3, the pore size of the soil was measured by an NMR test. The author should have explained in detail how the soil was tested in an NMR facility and the name of the software used.

7. The author should explain in the discussion section whether the higher degree of deterioration in the 145-155m upper level will lead to the occurrence of landslides?

8. The author should further simplify the conclusion of the paper.

9. The author also needs to polish the sentences of the paper.

Author Response

Response to Reviewer 1 Comments

Dear Editors and Reviewers:

Thank you for your letter and for the reviewers’ comments concerning our manuscript entitled “Study on soil deterioration mechanism in the subsidence zone of Wildcat Landslide in the Three Gorges Reservoir Area”(Manuscript ID: water-2144380). Those comments are all valuable and very helpful for revising and improving our paper, as well as the important guiding significance to our research. We have studied those comments carefully and have made corrections which we hope meet with approval. The main corrections responses paper and the responds to the reviewer’s comments are as flowing:

Point 1: The author should explain the purpose of this study more clearly in the abstract.

Response: Thank you for your precious comment. The details are modified as follows: “The stability of soil mass near the dam bank in the Three Gorges Reservoir is closely related to the periodic variation in the reservoir water level. In order to study the influence of water level variation on soil mass, the soil mass in the water level fluctuation zone of the Wildcat landslide was taken as the research object, and the mechanism of soil mass deterioration in this area was revealed by comparing the strength and mineral structure characteristics of soil mass at different elevations by means of macro- and meso-microscopic analysis.” Lastly, thank you again for your precious comments, they are very helpful to improve the readability and preciseness of our manuscript.

New lines: 1-6                                                          

Point 2: The author should introduce the innovation of his research content in more detail in the introduction.

Response: Thanks for your precious comment. This is useful for highlighting the innovative aspects of the article. The details are modified as follows: “At present, most scholars study the influence of reservoir water on soil by changing the test conditions of laboratory tests, but they ignore the difference in soil properties at different elevations caused by the change in water level in the fluctuation zone. Therefore, this paper focuses on an experimental test of soil mass in the water level zone of the Wildcat landslide and reveals the soil mass' deterioration mechanism by comparing its characteristics at different elevations and adopting the analysis method combining macro and meso.” Lastly, thank you again for your precious comments, they are very helpful to improve the readability of our manuscript.

New lines: 93-100

Point3: The calculation result in the table of 175m high level fractal dimension in paper 3.1 is inconsistent with that in the paper. Please check it carefully.

Response: Thanks very much for your valuable comments, we are very sorry for our carelessness, we have corrected the errors in the manuscript according to your suggestions, the details are modified as: “We have changed the incorrect data of 2.26 into the correct data of 2.15 for the fractal dimension of soil mass at elevation 175 in the Table 1.” Lastly, thank you again for your precious comments, they are very helpful for the rigor of our paper.

New lines: 186-187

Point 4: The author should explain the variation law of permeability coefficient and pore ratio in paper 3.1.

Response: Thanks for your comment, this is very useful to improve the depth of the paper. The details are modified as “According to the above rules, due to the periodic changes in reservoir water, the deterioration of the soil mass of 155 m is the most severe. The internal pores of the soil mass are large, and the water flow rate in the soil mass is fast. Therefore, the pore ratio and permeability coefficient of the soil mass at the height of 155 m are the largest.” Lastly, thank you again for your precious comments.

New lines: 217-220

Point 5: The authors should carefully check whether the pore size unit "μm" in paper 3.3 is misexpressed.

Response: Thanks for your comment, we are very sorry for our carelessness, we have corrected the errors in the manuscript according to your suggestions, the details are modified as: “We have modified the pore size unit "um" to "μm" in the text of Section 3.3 and Figure 9.” Lastly, thank you again for your precious comments. We will notice the writing standard carefully in the following works.

Figure 9. Pore size ratio of soil at different elevations

New lines: 290-291

Point 6: In paper 3.3, the pore size of the soil was measured by an NMR test. The author should have explained in detail how the soil was tested in an NMR facility and the name of the software used.

Response: Thanks for your valuable comment, it is quite helpful to improve our manuscript. The details are modified as: “According to the dry density of the soil mass, the corresponding volume mass was weighed, and the soil mass was pressed into a PVC pipe with a diameter of 50 mm and a height of 20 mm. The PVC pipe was used to prevent the interference of nuclear magnetic signals, and it was used in conjunction with the nuclear magnetic resonance testing equipment.” Lastly, thank you again for your precious comment, they are very helpful to improve the academic rigor of our study.

New lines: 283-288

Point 7: The author should explain in the discussion section whether the higher degree of deterioration in the 145-155m upper level will lead to the occurrence of landslides?

Response: Thank you for your valuable comments and suggestions, which are very helpful for my research ideas. We have the following explanation “In future studies, we will conduct a series of test studies on the soil at the height of 145-155m, including laboratory sample dry and wet cycle test, laboratory model test and numerical simulation, so as to analyze the probability of landslide caused by the soil in this region. Due to limited space, we will not give too much description this time.” Thank you again for pointing out the shortcomings in the research. We will strengthen the rigor of research in the following works.

Point 8: The author should further simplify the conclusion of the paper.

Response: Thanks for your valuable comment, it is quite helpful to improve our manuscript. The details are modified as: “The purpose of this study was to study the effect of periodic variation in water level on landslide soil. Taking the soil mass in the water level zone of Wildcat landslide as the research object, the deterioration mechanism of the soil mass in the water level zone of the Wildcat landslide was revealed through analysis of the macroscopic parameter variation rule and the meso-structural characteristics of the soil mass at different elevations. The main conclusions are as follows:

(1) The testing and analysis of the natural water content and dry density of soil at different elevations showed that the natural water content is negatively correlated with the sampling elevation. The dry density of the soil is at a minimum when the elevation is 155 m and at a maximum when the elevation is 175 m.

(2) Fractal theory was used to calculate the fractal dimension of soil at different elevations. When the elevation was 155 m, the fractal dimension of soil was the largest, and the soil particles were broken most severely. Combined with the strength characteristics of the soil mass, the soil mass can be divided by the deterioration degree at different elevations: deterioration enhancement, deterioration weakening, and no deterioration.

(3) A mineral test and nuclear magnetic resonance test conducted on the soil in different areas showed that, when the soil elevation was between 145 and 155 m, the amount of clay minerals and quartz in the soil was the lowest, and there were more internal pores in the soil.

(4) The periodic circulation of the water level will lead to the dissolution of clay minerals and quartz in the soil, and the number of internal pores in the soil will increase. Moreover, the periodic fluctuation of the water level will lead to the formation of seepage channels in the soil, and the macro strength of the soil will decrease.”. Thank you again for pointing out the shortcomings in the research.

New lines: 372-395

Point 9: The author also needs to polish the sentences of the paper

Response: Thanks for your precious comment. Thank you for pointing out that our research and expression are not rigorous. We polished the whole article in English. Thank you again for pointing out the shortcomings in the research. We will strengthen the rigor of research in the following works.

Specially, thanks to you for your good comments and suggestions again. We hope our corrections and explanations will meet your satisfaction. In all, we found the comments are quite helpful, and we revised our paper point-by-point. Thank you and the reviewers again for your help!

Yours

Sincerely,

Can Wei

Reviewer 2 Report

Dear editor;

In this study, the authors have studied the influence of periodic variation of reservoir water level on the macroscopic and mesoscopic characteristics of the soil mass on the bank slope of the reservoir. To achieve the main goal, the authors have conducted a series of indoor tests on the macroscopic parameters and mesoscopic structure of the soil mass at different elevations in the water-leveling zone of Wildcat landslide. The overall structure of the paper is okay. However, there are some comments for possibly improving the manuscript.

- Abstract needs to modify: the abstract should contain Objectives, Methods/Analysis, Findings, and Novelty/Improvement.

- Originality/novelty of the study proposed. This issue is very important and should be better clarified and well-highlighted in the both abstract and introduction.

-This raises some concerns regarding the potential overlap with authors previous works. The authors should explicitly state the novel contribution of this work, the similarities and the differences of this work with their previous publications.

- For readers to quickly catch your contribution, it would be better to highlight major difficulties and challenges, and your original achievements to overcome them, in a clearer way in abstract and introduction.

-Footnote in Table 1 is not clear.

-Scale Drawing and north direction should be added in Figure 1.

- Materials and methods is maybe the most important heading to evaluate the general quality of any research dissemination product, since it is the one that explains to the readers what procedures, approaches, methods, datasets, methodology, and treatments used in the paper. Unfortunately, the manuscript did not pay sufficient attention to this element.

-The reference format is not consistent with the journal’s requirement (For example, there is no point between Abbreviated Journal Name and Year). The authors should carefully check through the editorial issues before resubmission.

-At the methodology, the authors are suggested to add a flowchart of the overall research, and then introduce each part of the flowchart, which makes the readers catch the key point easily.

- The discussion section needs to be described scientifically. Kindly frame it along the following lines: i. Main findings of the present study ii. Comparison with other studies iii. Implication and explanation of findings iv. Strengths and limitations v. Conclusion, recommandation, and future direction.

Author Response

Response to Reviewer 2 Comments

Dear Editors and Reviewers:

Thank you for your letter and for the reviewers’ comments concerning our manuscript entitled “Study on soil deterioration mechanism in the subsidence zone of Wildcat Landslide in the Three Gorges Reservoir Area”(Manuscript ID: water-2144380). Those comments are all valuable and very helpful for revising and improving our paper, as well as the important guiding significance to our research. We have studied those comments carefully and have made corrections which we hope meet with approval. The main corrections responses paper and the responds to the reviewer’s comments are as flowing:

Point 1: Abstract needs to modify: the abstract should contain Objectives, Methods/Analysis, Findings, and Novelty/Improvement.

Response: Thank you for your precious comment. it’s very helpful to improve the readability of our manuscripts. We have revised the abstract and the details are modified as follows: “The stability of soil mass near the dam bank in the Three Gorges Reservoir is closely related to the periodic variation in the reservoir water level. In order to study the influence of water level variation on soil mass, the soil mass in the water level fluctuation zone of the Wildcat landslide was taken as the research object, and the mechanism of soil mass deterioration in this area was revealed by comparing the strength and mineral structure characteristics of soil mass at different elevations by means of macro- and meso-microscopic analysis.” Lastly, thank you again for your precious comments, it is very helpful to improve the readability and preciseness of our manuscript.

New lines: 1-6                                                          

Point 2 Originality/novelty of the study proposed. This issue is very important and should be better clarified and well-highlighted in the both abstract and introduction.

Point 3 This raises some concerns regarding the potential overlap with authors previous works. The authors should explicitly state the novel contribution of this work, the similarities and the differences of this work with their previous publications.

Point 4 For readers to quickly catch your contribution, it would be better to highlight major difficulties and challenges, and your original achievements to overcome them, in a clearer way in abstract and introduction.

Response2-4: Thanks for your precious comment. The introduction of the manuscript has been greatly modified. We have revised the introduction and the details are modified as follows: “At present, most scholars study the influence of reservoir water on soil by changing the test conditions of laboratory tests, but they ignore the difference in soil properties at different elevations caused by the change in water level in the fluctuation zone. Therefore, this paper focuses on an experimental test of soil mass in the water level zone of the Wildcat landslide and reveals the soil mass' deterioration mechanism by comparing its characteristics at different elevations and adopting the analysis method combining macro and meso”. Lastly, thank you again for your precious comment. We will strengthen the rigor of research in the following works.

New lines: 93-100

Point 5: Footnote in Table 1 is not clear.

Response: Thanks for your comment, we are very sorry for our carelessness, this paper table does not have footnote, we have removed it. We will notice the writing standard carefully in the following works.

New lines: 186-187

Point 6: Scale Drawing and north direction should be added in Figure 1.

Response: Thanks for your valuable comment, it is quite helpful to improve our manuscript. the details are modified as follows: We added Scale Drawing and north direction in Figure 1 of the paper. Lastly, thank you again for your precious comment, it is very helpful to improve the academic rigor of our study.

Figure 1. Sampling spot

New lines:116-117

Point 7: Materials and methods is maybe the most important heading to evaluate the general quality of any research dissemination product, since it is the one that explains to the readers what procedures, approaches, methods, datasets, methodology, and treatments used in the paper. Unfortunately, the manuscript did not pay sufficient attention to this element.

Point 9: At the methodology, the authors are suggested to add a flowchart of the overall research, and then introduce each part of the flowchart, which makes the readers catch the key point easily.

Response 7 and 9: Thank you for your valuable comments and suggestions, which are very helpful for my research ideas. We have revised the materials and methods and the details are modified as follows: “The test process of this study is shown in Figure 2. (1) When the lowest water level of the Three Gorges Reservoir area was 145 m, soil samples were selected from the Wildcat landslide site at different elevations. The elevations were 145 m, 155 m, 165 m, 175 m, and 180 m, and 50 kg soil samples were selected for each elevation to prevent the loss of water in the soil. (2) After the soil sample was retrieved, the wax sealing method was used to test the dry density of the soil in strict accordance with the specification GB/T50123-1999 Geotechnical Test Method, as shown in Figure 2 (A). At the same time, the remaining samples were naturally dried. After drying, the dry soil sample was hammered out with a rubber hammer. (3) Screening tests were conducted on the soil at different elevations, as shown in Figure 2 (B). After screening, the constant head method was used to test the soil permeability coefficient, as shown in Figure 2 (C). Direct shear tests were conducted on soil samples to test the shear strength of soil, as shown in Figure 2 (D). (4) X-ray diffraction (XRD) was used to test the mineral composition of soil at different elevations and observe its mineral composition and content, as shown in Figure 2 (a). (5) Nuclear magnetic resonance (NMR) was used to test the pore characteristics of soil at different elevations, as shown in Figure 2 (c). Because the periodic change in water level leads to the different characteristics of soil at different elevations, the mechanism of soil deterioration in the subsidence zone can be revealed by using the analysis method combining macro and meso.”. Thank you again for pointing out the shortcomings in the research. We will strengthen the rigor of research in the following works.

New lines: 113-130

Point 8: The reference format is not consistent with the journal’s requirement (For example, there is no point between Abbreviated Journal Name and Year). The authors should carefully check through the editorial issues before resubmission.

Response: Thanks for your precious comment and question. We are very sorry for our carelessness, in strict accordance with the requirements of the journal, we have changed the format of the paper reference and deleted the points in the journal name and year. Thanks again for your valuable comments, we will strengthen the rigor of research in the following works.

New lines: 408-464

Point 10: The discussion section needs to be described scientifically. Kindly frame it along the following lines: i. Main findings of the present study ii. Comparison with other studies iii. Implication and explanation of findings iv. Strengths and limitations v. Conclusion, recommandation, and future direction.

Response: Thanks for your precious comment and question. which are very helpful for my research ideas. We have revised the materials and methods and the details are modified as follows:” Through the combination of macro and meso test analysis, it is found that the main reason for the different deterioration degrees of soil at different elevations is that soil at different elevations experiences different degrees of water-soil physical and chemical effects. The closer the elevation is to the reservoir water level, the stronger the physical and chemical effects are experienced. Therefore, it is of great significance for landslide prevention and prediction to fully understand the macroscopic mechanical characteristics and meso-structural characteristics of soil mass in the subsidence zone. This study focuses on soil mass of different elevations in the same region. In the future, the research group will compare soil mass of the same elevation in different regions to study the deformation and failure mechanism of soil mass, and to reveal the relationship between soil mass failure in the subsidence zone and the deformation and instability of the landslide body.” Lastly, thank you again for your precious comments, it is very helpful to improve the readability and preciseness of our manuscript.

New lines: 361-371

Special thanks to you for your good comments and suggestions again. We hope our corrections and explanations will meet your satisfaction. In all, we found the comments are quite helpful, and we revised our paper point-by-point. Thank you and the reviewers again for your help!

Yours

Sincerely,

Can Wei

Reviewer 3 Report

By investigating the periodic change of water level in the Three Gorges Reservoir, the authors studied the degradation of soil on the bank slope of the reservoir. This study took soil at different elevations in Wildcat region as the research object. By testing soil parameter changes and mesostructure characteristics at different elevations, the difference of soil quality degradation in Wildcat region and its mechanism were revealed. It is of great significance to evaluate and forecast geological hazards in similar regions. But the content of the paper has some defects, hope the following suggestions can be helpful to the improvement of the paper:

(1) In Section 2 Materials and Methods, The author mentions "The reasons for the difference in macroscopic parameters of soil at different elevations were analyzed and the deterioration mechanism of soil in the zone was revealed." This section describes the method to get the mechanism or cause. It does not analyze the reason for the difference of soil macro parameters at different elevations, nor can it get the conclusion that reveals the mechanism of soil deterioration.

(2) Some of the pictures in the text are placed incorrectly. The pictures should be placed after the body paragraph description or after the relevant paragraph.

(3) Equation 5 can be derived from the combination of Equation 2 and Equation 3. Is it need to integrate Equation 4?

(4) The fractal dimension D of soil in the paper is between 2.15 and 2.28, while the fractal dimension of soil in Table 1 is between 2.20 and 2.28. Which is correct?

(5) In The last sentence of the last paragraph of Section 3.1, "The elevation of region III is more than 175m, and the permeability is weak and strengthened." It should be said that "permeability is lowest at 175m, but gradually increases with elevations".

(6) Lines 229-230 are repeated

Author Response

Response to Reviewer 3 Comments

Dear Editors and Reviewers:

Thank you for your letter and for the reviewers’ comments concerning our manuscript entitled “Study on soil deterioration mechanism in the subsidence zone of Wildcat Landslide in the Three Gorges Reservoir Area”(Manuscript ID: water-2144380). Those comments are all valuable and very helpful for revising and improving our paper, as well as the important guiding significance to our research. We have studied those comments carefully and have made corrections which we hope meet with approval. The main corrections responses paper and the responds to the reviewer’s comments are as flowing:

Point 1: In Section 2 Materials and Methods, The author mentions "The reasons for the difference in macroscopic parameters of soil at different elevations were analyzed and the deterioration mechanism of soil in the zone was revealed." This section describes the method to get the mechanism or cause. It does not analyze the reason for the difference of soil macro parameters at different elevations, nor can it get the conclusion that reveals the mechanism of soil deterioration.

Response: Thanks very much for your valuable comments, which are helpful for us to improve quality and readability of the manuscript. The details are modified as follows: “Because the periodic change in water level leads to the different characteristics of soil at different elevations, the mechanism of soil deterioration in the subsidence zone can be revealed by using the analysis method combining macro and meso.” Lastly, thank you again for your precious comments, it is very helpful to improve the readability and preciseness of our manuscript.

New lines: 127-130                                                  

Point 2: Some of the pictures in the text are placed incorrectly. The pictures should be placed after the body paragraph description or after the relevant paragraph.

Response: Thanks very much for your valuable comments, which are helpful for us to improve readability of the manuscript. We have placed figures 1 and 2 from the paper in the Methods and Materials section of the paper. Lastly, thank you again for your precious comment. It is very helpful to improve the readability of our manuscript.

Point 3: Equation 5 can be derived from the combination of Equation 2 and Equation 3. Is it need to integrate Equation 4.

Response: Thanks very much for your valuable comments, which are helpful for us to improve quality and readability of the manuscript. “In the third section of the paper, we explained that formula (5) needs to integrate formula (4).” Lastly, thank you again for your precious comments, it is very helpful to improve the readability of our study.

New lines:170-176

Point 4: The fractal dimension D of soil in the paper is between 2.15 and 2.28, while the fractal dimension of soil in Table 1 is between 2.20 and 2.28. Which is correct.  

Response: Thanks very much for your valuable comments, we are very sorry for our carelessness, we have corrected the errors in the manuscript according to your suggestions, the details are modified as: “We have changed the incorrect data of 2.26 into the correct data of 2.15 for the fractal dimension of soil mass at elevation 175 in the Table 1.” Lastly, thank you again for your precious comments, it is very helpful for the rigor of our paper.

New lines: 186-187

Point 5: In The last sentence of the last paragraph of Section 3.1, "The elevation of region III is more than 175m, and the permeability is weak and strengthened." It should be said that "permeability is lowest at 175m, but gradually increases with elevations".

Response: Thanks for your comment, we have corrected the resolution of Fig 7 to clearly see the length of the scale, and indicated in the manuscript that the scale length is 1 μ m. The details are modified as “The elevation of region III is more than 175m, permeability is lowest at 175m, but gradually increases with elevations.” Lastly, thank you again for your precious comments, it is very helpful to improve the readability and preciseness of our manuscript.

New lines: 216

Point 6: Lines 229-230 are repeated.

Response: Thanks very much for your valuable comments, we are very sorry for our carelessness. We have deleted the repetitive parts in the paper. Lastly, thank you again for your precious comment, it is very helpful to improve the academic rigor of our study.

New lines:232-233

Special thanks to you for your good comments and suggestions again. We hope our corrections and explanations will meet your satisfaction. In all, we found the comments are quite helpful, and we revised our paper point-by-point. Thank you and the reviewers again for your help!

Yours

Sincerely,

Can Wei

Round 2

Reviewer 3 Report

Good!